# Description of a New miRNA Signature for the Surgical Management of Thyroid Nodules

**DOI:** 10.3390/cancers16244214

**Published:** 2024-12-18

**Authors:** Marie Quiriny, Joel Rodrigues Vitόria, Manuel Saiselet, Geneviève Dom, Nicolas De Saint Aubain, Esther Willemse, Antoine Digonnet, Didier Dequanter, Alexandra Rodriguez, Guy Andry, Vincent Detours, Carine Maenhaut

**Affiliations:** 1Institut Jules Bordet, HUB, Université libre de Bruxelles, 1070 Brussels, Belgium; n.desaintaubain@hubruxelles.be (N.D.S.A.); esther.willemse@hubruxelles.be (E.W.); antoine.digonnet@hubruxelles.be (A.D.); guy.andry@hubruxelles.be (G.A.); 2IRIBHM Jacques E. Dumont, Université libre de Bruxelles, 1070 Brussels, Belgium; joel.rodrigues.vitoria@ulb.be (J.R.V.); manuel.saiselet@gmail.com (M.S.); genevieve.dom@ulb.be (G.D.); vincent.detours@ulb.be (V.D.); carine.maenhaut@ulb.be (C.M.); 3Department of Otolaryngology-Head and Neck Surgery, CHU Saint-Pierre, Université Libre de Bruxelles, 1070 Brussels, Belgium; didier.dequanter@pandora.be (D.D.); alexandra.rodriguez@stpierre-bru.be (A.R.)

**Keywords:** microRNA, molecular signature, thyroid carcinoma, thyroidectomy, fine-needle aspiration biopsy

## Abstract

We present a new molecular signature, based on altered miRNA expressions and specific mutations, allowing for improving the screening of malignant thyroid nodules. This is a prospective non-interventional study, including all Bethesda categories, carried out on an FNAB sampled in suspicious nodule(s) during thyroidectomy. The reference diagnosis was the pathological assessment of the surgical specimen. miRNA quantification and mutations detection were performed. Different classification algorithms were trained with molecular data to correctly classify the samples. The random forest was the best algorithm. This classifier used mostly miRNAs to classify the nodules. This classifier is able to identify malignant nodules with a high PPV and NPV (both 90%), a high specificity (96%) and a competitive sensitivity (76%). Our data suggest that miRNA expressions emerge as more reliable first-line diagnostic markers. This signature could be efficient to improve the screening of thyroid cancer as a complementary test in clinical practice, to reduce the rate of unnecessary surgery.

## 1. Introduction

Follicular-thyroid-cell-derived tumors include benign and malignant forms with distinct histological presentation. Thyroid cancers are essentially papillary carcinomas (about 80%) and follicular carcinomas (about 15%). Most of these cancers are associated with genetic abnormalities that alter the MAP Kinase (MAPK) or the PI3Kinase pathways. In papillary thyroid carcinoma, the most frequent genetic alterations are the BRAF^V600E^-activating mutations (62%) and RAS-activating mutations (13%). RET/PTC chromosomal rearrangements are found in approximately 6% [1].

One of the major public health problems of thyroid tumors is the high rate of thyroid nodules discovered by ultrasonography (US) (67%) [2]. The rate of incidental thyroid nodules by US ranges from 30 to 50%, and 5 to 15% are malignant. Recently, their incidence has increased [2]. The criteria to identify a suspicious nodule are defined in the European Thyroid Imaging and Reporting Data System (EU-TIRADS) [3]. The diagnosis of malignant thyroid nodules is based on the cytological results of the fine-needle aspiration biopsy (FNAB). This procedure is carried out under ultrasonography. A total of 5.8% of FNABs are inadequate, and 11.2% are indeterminate [4,5]. The six general diagnostic categories of the FNAB are classified in the Bethesda System [6]. The re-examination of the nodule with a repeated FNAB is often requested for the unsatisfactory (Bethesda I) and the undetermined (Bethesda III) samples. The rate of malignancy among these samplings (Bethesda I and III) is about 20 to 25% [5]. For atypia of undetermined significance (AUS) or for follicular lesion of undetermined significance (FLUS) (Bethesda III classification), repeated FNAB, molecular testing or lobectomy are the usual practices. For follicular neoplasm or for suspected follicular neoplasm (Bethesda IV classification), the usual management is a lobectomy or a molecular testing to obtain a more specific risk factor. For the Bethesda V category, there is a 45 to 60% rate of suspicion of malignancy, and the surgical management is near-total thyroidectomy or lobectomy.

Thyroidectomy may be associated with complications, and the consequence of this intervention is hormonal substitution for life [7,8]. To prevent unnecessary surgery and improve the detection of malignant nodules, different molecular tests have been developed, mainly based on mutation detection and on mRNA or miRNA expression. However, these tests are not yet sufficiently precise to reduce drastically the number of unneeded thyroidectomies. Therefore, it is important to find other molecular markers for screening.

MicroRNAs (miRNA) are small non-coding RNAs, 19 to 25 nucleotides long, which repress the stability and the translation efficiency of mRNAs. Their expression is deregulated in many cancers, including thyroid cancer, and experimented research has revealed the role of several miRNAs in cancer pathogenesis, acting as oncogenes or tumor suppressors [9]. Papillary thyroid carcinoma (PTC) and follicular thyroid carcinoma (FTC) present up- and downregulated miRNAs, while anaplastic thyroid carcinoma (ATC) shows almost exclusively downregulated miRNAs [10].

In a previous work, we analyzed miRNA expression in primary PTC and associated nodal metastases. New modulations of miRNA expressions were identified, appearing as potential biomarkers for thyroid cancer [11]. From these data and our review of the literature [9], we devised a new molecular classifier that combines miRNA expression analysis with mutation detection. This study was undertaken to assess the performance of this signature with the aim to improve the detection of malignant thyroid nodules.

## 2. Materials and Methods

### 2.1. Sample Collection

This is a prospective non-interventional study. From 2015 to 2019, we collected 312 per-operative samples from patients (≥18 years of age) in two centers in Brussels: the Jules Bordet Institute and the University Central Hospital Saint-Pierre. The surgical decision was based on the surgical guidelines for adult patients with thyroid nodules and differentiated thyroid cancer [1]. The recorded clinical data were age, sex, cancer type, staging of the tumor and Bethesda classification of the preoperative FNAB.

All patients had a preoperative US and were staged according to the TIRADS classification [3]. Most of them had a preoperative FNAB. The preoperative FNA samples were dispersed in Cytorich liquid and stained with Papanicolaou. The cytological technique used for the cytological assessment was the liquid preparation of Surepath^®^. The samples were classified using the Bethesda classification [6]. Some patients had surgical indication (size of the nodules > 4cm, progression of thyroid nodules…) without preoperative FNAB. No patients had a preoperative molecular test. The reference diagnosis was based on the pathological assessment of each surgical specimen. These operative specimens were fixed in 4% formalin and embedded in paraffin. Five µm thick sections, stained with hematoxylin and eosin (H&E), were reviewed by the same pathologist for both centers. All FNABs had a traceable surgical pathology outcome with a documented histological diagnosis. During the intervention, the surgeon performed an FNAB in the suspicious nodule(s) of the surgical specimen ex vivo. These nodules corresponded to the same suspicious nodules identified by the preoperative US. Patients with small nodules (<1 cm) that could not be identified during surgery were excluded. The samples were collected in an RNA conservative solution (Qiazol, Qiagen, Hilden, Germany), homogenized, and stored at −80 °C. The study was performed following the double-blinded rules, to ensure unbiased data generation and classification: pathologists ignored the molecular analysis results, and none of the personnel involved in the molecular analyses were aware of histopathological classifications. The study was approved by the local ethical committees of both collecting centers, and samples were anonymized.

### 2.2. RNA Extraction, Quality Controls and Characteristics of Eligible Samples

One µL of UniSp2, UniSp4 and UniSp5 RNA Spike-in template mix (Qiagen, Hilden, Germany) was added to each sample, and total RNA was extracted and purified following miRNeasy micro kit recommendations (Qiagen, Hilden, Germany). The quality controls performed are described in the Appendix A.

### 2.3. Molecular Analyses

The molecular analyses were established after literature review about the role of miRNAs and oncogenic gene mutations in thyroid cancer [12,13,14,15]. A total of 36 candidate miRNA biomarkers were selected from our previous study [9,10,11]: hsa-miR-1179, hsa-miR-7-2-3p, hsa-miR-7-5p, hsa-miR-204-5p, hsa-miR-139-5p, hsa-miR-451a, hsa-miR-152-3p, hsa-miR-375, hsa-miR-873-5p, hsa-miR-23a-3p, hsa-miR-146b-5p, hsa-miR-155-5p, hsa-miR-221-3p, hsa-miR-222-3p, hsa-let-7a-5p, hsa-miR-182-5p, hsa-miR-183-5p, hsa-miR-126-5p, hsa-miR-484, hsa-miR-148b-3p, hsa-miR-125a-3p, hsa-miR-199a-5p, hsa-miR-125b-5p, hsa-miR-138-1-3p, hsa-miR-551b-3p, hsa-miR-31-5p, hsa-miR-30d-5p, hsa-miR-9-5p, hsa-miR-21-5p, hsa-miR-150-5p, hsa-miR-23b-3p, hsa-miR-223-3p, hsa-miR-151a-3p, hsa-miR-197-3p, hsa-miR-99a-5p, and hsa-miR-214-3p. Quantifications were performed following miRCURY LNA kit recommendations (Exiqon/Qiagen, Hilden, Germany) by reverse transcription and quantitative PCR. Ten ng of total RNA of each sample were tested in 96-well plates containing miRCURY LNA primer sets (Qiagen, Hilden, Germany) specific for each target miRNA and spike-in (three quality controls and one inter-plate independent calibrator per sample). Non-RNA and non-cDNA template conditions were run after every 10 samples to guarantee the absence of contamination throughout the process. PCR amplifications were performed on a CFX96 Touch Real-Time PCR Detection System (Bio-Rad, Hercules, CA, USA) for 40 cycles (10 s at 95 °C and 60 s at 56 °C) followed by a melting curve analysis. Expression values were obtained for each miRNA in each sample using the CFX Maestro Software (Bio-Rad, Hercules, CA, USA).

Point mutations in the BRAF (codon 600), HRAS (codon 12 and 61), KRAS (codon 12 and 13) and NRAS (codon 12 and 61) genes were detected in addition to RET/PTC1, RET/PTC2 and PAX8/PPARG gene fusions (detailed in Appendix A).

### 2.4. Data Analyses

In order to streamline the choice of classification algorithm and training approach, we used the R statistical software version 4.1.0 [16] and built it on the Caret package, version 6.0.92 [17]. A first prototype using a preliminary subset of the FNAB samples was used to test 8 algorithm candidates implemented in the Caret package: random forest, Support Vector Machines with Linear Kernel and with Radial Basis Function Kernel, Multi-Layer Perceptron with multiple layers, Neural Network, C5.0 and Regularized Logistic Regression. Using ROC curves and predictive metrics, later iterations of the pipeline settled on using the 3 best candidates: random forest, Regularized Logistic Regression, Support Vector Machines with Linear Kernel. The raw input data were first cleaned (removing samples and miRNAs flagged as problematic due to bad quality, technical problems and/or high number of missing values, e.g., miRNAs “hsa-miR-876-5p”, ”hsa-miR-129-5p”, ”hsa-miR-187-3p”, ”hsa-miR-204-3p”, ”hsa-miR-625-3p”). The remaining missing values (1.85% of the total sample points) were manually set to the value of 40 Ct, which we consider to be the maximal theoretical detection value for the qPCR. We applied the preprocess function of the Caret package for centering and scaling of both datasets. Instead of using reference genes for our PCR value normalizations, we applied a “ratio” approach of all the tested miRNAs. In practice, we calculated each unique delta for the miRNA C_t_ values (36 miRNAs, 630 unique deltas combinations). We then added a status column (malignant or benign) and 7 additional columns for each possible mutation status (no mutations, BRAF, RET-PTC1, RET-PTC3, NRAS, HRAS, KRAS and PAX8-PPARG), leading to a total of 638 value columns. For further analysis, we split the processed dataset (294 samples) into two subsets—training set (147 samples) and the final validation set (147 samples)—following a 50/50 ratio of malign/benign samples in each subset. The training was only applied on the training set using a repeated cross-validation approach using 20 multi-folds of 5 subsets each. The space of possible parameters for each prediction algorithm was searched randomly using the tunelength parameter (i.e., 50) in the Caret package. The best parameters and models were selected based on the best accuracy score. The validation set was then used to independently test the retained models. For this, we predicted each sample, creating 2 by 2 contingency tables and prediction metrics (sensitivity, specificity, precision, recall, F1, positive and negative predictive value, accuracy, and kappa values). We also plotted the prevalence versus the positive and negative predictive values (using observed sensitivity and specificity) to evaluate each model at different prevalence points. Based on similar studies, we considered the non-invasive follicular thyroid neoplasms with papillary-like nuclear features (NIFTP) as malignant nodules requiring a surgical excision [18]. Observed sensitivity and specificity were used to calculate hypothetical post-test predictive values over the entire range of possible prevalence of the disease using the Bayes theorem. Statistical analyses were performed in R.

### 2.5. Analytical Validation

Minimal acceptable tumor cell content, minimal acceptable thyroid cell content and assay precision regarding the validity of our miRNA classifier results were determined (Appendix A).

## 3. Results

In a previous study [11], by performing miRNA deep sequencing, we found new modulations of miRNA expressions in papillary thyroid tumors and nodal metastases. We also identified downregulated miRNAs, representing new potential biomarkers for thyroid cancer. From these data and our review of the literature [9], we developed a new molecular classifier based on the expression of 36 miRNA and on the detection of principal mutations. We chose commonly reported differentially expressed miRNA that were modulated in most of the expression profile studies available at that time. We selected miRNA modulated in PTC and FTC.

To evaluate the performance of our signature, 276 patients requiring thyroidectomy were included in the study (Figure 1). Preoperative FNAB was performed in 312 suspicious nodules (nodules < 1 cm excluded). RNA was purified from those samples (seven were excluded because of RNA degradation). PAX8 mRNA expression was detected in each of the 305 samples; however, spike-in quantification revealed that 4 RNA extractions were not satisfactory, and those samples were excluded. Because of a lack of surgical data or histological results, seven additional samples were rejected. The remaining 294 FNABs were used for miRNA profiling and classical mutation analyses (Appendix A).

These 294 samples were randomly divided in two equal groups of 147 samples, constituting the training and the validation set. The primary goal of separating the data into training and validation sets was to ensure the robustness and generalizability of the predictive model. The training set was used to develop the model by allowing the machine learning algorithms to learn patterns and relationships in the data. The validation set, an independent subset, was used to assess the model performance on unseen data, providing an unbiased estimate of how the model will perform in real-world applications. To ensure a representative distribution of benign and malignant cases in both sets, we stratified the samples by their clinical outcome (malignant vs. benign) before random assignment to the training and validation sets. This step minimizes potential biases and ensures that both subsets reflect the overall dataset characteristics. These characteristics are summarized in Table 1. No patients had metastatic lymph nodes on preoperative ultrasonography. No significant difference was found between the groups regarding the disease prevalence (41.84%) and the clinical characteristics. The median age of the cohort, composed of 75% women, was 52 years. We included nodules classified as THY 1 to 6 according to the Bethesda classification. Among the malignant histological results, papillary thyroid cancer was the most frequent diagnosis. These randomized groups were thus comparable.

Following miRNA profiling and classical mutation analyses, we computed all possible pairwise differences in logCt values between miRNAs (e.g., logCt(miRNA1)—logCt(miRNA2)) to generate the input features for our machine learning model. This approach avoids the reliance on a single reference gene or external control for normalization and instead focuses on relative expression levels between miRNAs. We believe this strategy allows the model to capture biologically relevant patterns while being robust to potential biases introduced by normalization to a single control. Using the training set, we compared the accuracy of eight different algorithms to detect malignant nodules. Indeed, our primary objective was to identify a predictive model that achieved the highest accuracy, as this was the metric specified during training via the Caret package in R. Using repeated cross-validation (5-fold, repeated 20 times) on the training set, we optimized multiple algorithms and selected the model with the highest accuracy for further validation (Figure 2A). The random forest (rf) algorithm presented this highest accuracy (AUC = 0.8578) and was thus considered as the best one (Figure 2B). The statistical power of our classifier was therefore derived from the results obtained by the rf algorithm.

Figure 3 presents the 25 best molecular markers used by random forest, ranked by the relative importance score attributed by the algorithm. It summarizes which miRNA delta pairs were the most important to predict the outcome (benign vs. malignant). Boxplots of the raw Ct values for the 36 miRNAs are presented in Appendix A, and the complete list of molecular markers is provided in Appendix A. The algorithm used mostly miRNAs to classify the nodules. The mutational status, and particularly the BRAF^V600E^ mutation, was not among the 25 best markers, and RAS mutation was not retained by any classification algorithm.

The statistical power of our classifier was calculated from the validation set (66 malignant and 81 benign nodules) (Table 2). Among the benign nodules, random forest (rf) identified 3 as malignant and 78 as benign. Among the malignant nodules, 50 were classified as malignant and 16 as benign. The sensitivity and the specificity were, respectively, 76% and 96%, and the PPV and the NPV were, respectively, 94% and 83%, at a prevalence of malignancy around 42%. These predictive values vary according to the disease prevalence, and their relationship is presented in Figure 4. Considering a malignancy prevalence of 30% as in relevant studies, the sensitivity and the specificity of our signature were, respectively, 76% and 96%, and the positive and negative predictive values were both 90%.

## 4. Discussion

In this study, we investigated the performance and potential clinical utility of a new molecular classifier to improve the risk stratification of thyroid nodules, based on miRNA quantification and mutation detection.

Thyroid nodules are commonly encountered; most of them are benign but 5% [2] are diagnosed as malignant. There is a need to distinguish them from the benign nodules in order to limit unnecessary surgery (lobectomy or near-total thyroidectomy). Cancer screening is based on FNAB [2] and classified following the Bethesda System [6]. Three diagnostic categories (I, III, and IV) are uncertain and associated with unnecessary surgery. The search for biomarkers in FNABs could improve the accuracy of the screening [13], but to date, the performances of the existing tests are often not sufficient for routine use, although they can help resolve diagnostic challenges. In addition, they are expensive. A confirmation test for the detection of cancer should have a high NPV and a high specificity. To prevent unnecessary surgery, a high PPV is also required. Our molecular classifier was built based on our previous study and on our review of the literature data [9,11]. It relies on the expression levels of 36 miRNAs and on the detection of 6 genetic alterations (BRAF^V600E^, NRAS, HRAS mutations, RET/PTC1, RET/PTC3, PAX8/PPARG fusions). Some of the miRNAs of our signature are present in the existing commercial tests, but we have included additional miRNAs for a more efficient classifier, which achieved an NPV of 83%, a PPV of 94%, a sensitivity of 76% and a specificity of 96% (for a disease prevalence of 42%). Predictive values of molecular classifiers depend on the prevalence of malignancy. Using the Bayes theorem, we extrapolated our results for a malignancy prevalence of 30% (Figure 4) to compare to the main molecular tests described in the literature. In these conditions, the NPV and the PPV were both 90%. All Bethesda categories were included in our study, so for potential clinical use, we would need to increase the number of Bethesda III and IV samples to reach a sufficient number in order to evaluate our signature specifically on these categories.

Multiple commercially available molecular tests are currently used to guide clinical management. Table 3 shows a comparison of the performance of our classifier with that of recent molecular tests, for a malignancy prevalence around 30%. Under this condition, our molecular test showed an NPV and PPV of 90%. The diagnostic accuracy of the ThyroSeq v3, which analyses mutations, gene fusions, copy number and gene expression alterations [19], was evaluated on Bethesda III, IV and V samples. Its sensitivity (93%) and NPV (97%) were higher but its specificity (81%) and PPV (68%) were lower than our classifier. The Rosetta GX Reveal, a miRNAs-based assay [20], included Bethesda II to VI samples. With its low PPV (59%), this test appears to be less efficient in order to prevent unnecessary thyroidectomies. The Multiplatform mutation and miRNA test [21] was evaluated on a small cohort (109 samples), composed of Bethesda III and IV samples. It presents a high NPV of 94% but a PPV of 74%, which is lower than the PPV of our classifier. Contrary to our work, in the study [21], miRNA expression levels were measured secondarily, after mutation detection. The Multiplatfom mutation and miRNA test operates like a “decision tree” and analyzes only 10 miRNAs [21]. This test resulted in ThyGenX-ThyraMIR. Finally, the Mir-THYpe, a miRNA-based classifier [22], was evaluated on the largest cohort (440 samples) and included a small number of resected test-negative nodules. Therefore, the authors performed a theoretical calculation based on the Bayes theorem and adjusted the sensitivity and specificity. For a disease prevalence around 30%, the PPV (66%) was much lower than the PPV of our classifier (90%) while the NPV was higher (95%).

When compared to these different molecular tests, our classifier has the highest specificity and PPV, although we are aware that these parameters may change if we focus exclusively on the Bethesda III and IV categories. This will be evaluated in the future. Its sensitivity is low (76%), but we must consider it as complementary to the Bethesda classification and possibly to other screening tests. Consequently, it could be used as a second line test to discriminate the indeterminate FNABs. For a disease prevalence around 30%, our test provided a competitive NPV of 90%. Using the Bayes theorem, for a disease prevalence of 20%, the NPV and PPV achieved 94% and 84%, respectively, suggesting that our signature could be useful, in theory, to prevent unnecessary surgeries.

Among the different histopathologic types of thyroid nodules, NIFTP constitutes a challenging diagnostic category. In our study, we recorded less NIFTP (2%, Table 1) in comparison with the studies of Steward (4%) [23], Lithwick-Yanai (7–8%) [20] and Santos (9%) [22]. This number was not specified in the study of Labourier [21]. Most NIFTPs are classified Bethesda III, IV or V. Usually, classical PTCs are well distinguished from NIFTP by FNAB, unlike follicular lesions. According to the non-cancer designation of NIFTPs, surgery remains the treatment but this could be more conservative [18]. All NIFTPs were classified as malignant by our molecular classifier, similarly to the ThyroSeq v3 study [19]. Rosetta GX Reveal [20] classified 36% of NIFTPs as benign in one set and all as malignant in another group. In the work of Santos [22], all NIFTPs were classified as malignant by the mir-THYpe assay. These discordant results reflect the high level of inter-observer variability in the diagnosis of this histological type. The misclassification of NIFTPs by the different tests could have surgical implications (lobectomy versus thyroidectomy) because their management changed [18,24]. Considering a larger cohort of NIFTPs to develop a diagnostic test for their specific detection would be beneficial.

An ongoing challenge is the preoperative distinction of benign follicular adenoma (FTA) from FTC, the second most common type of thyroid cancer (10–20%), which are very similar both histologically and genetically. The follicular lesions are classified as Bethesda III or IV. Their microscopic features are similar, and these lesions are not distinguishable by ultrasonography [24]. The diagnosis of FTC is based on capsular and/or vascular invasion, extra-thyroidal tumor extension, lymph node or systemic metastases. Hence, frozen section examination is not helpful to confirm the diagnosis of FTC. In this work, we included in the training and in the validation sets, respectively, 10 (7%) and 7 (5%) FTCs. Half of them were identified in each group by our molecular classifier, suggesting that it is not efficient for the detection of FTC. By comparison, 2 to 6% of FTCs were included in the other studies and 67 to 80% as malignant, but 84% of their FTAs were also classified as malignant, representing a high number of false positives [22]. As we reported previously, most deregulated mRNAs were common between FTA and FTC; however, FTC showed additional deregulated mRNA [25]. No mRNA signature capable of adequately classifying them was found, supporting the notion of continuous evolving tumors, displaying quantitative rather than qualitative changes. Similar results were obtained for miRNA expression, by our own and other groups [25,26], supporting that most FTCs derive from FTAs. Deregulated miRNAs are far less numerous in FTC than in PTC, and very few are common between different studies [26,27]. Some miRNAs have, however, been reported as specifically deregulated in FTC: for instance, miR-146a expression is reduced only in widely invasive FTC [27], and miR-129-1-3p is upregulated in FTC (fold change: 2.85) compared to FTA [25]. In the current study, miR-129-1-3p was not part of the miRNA signature; it could partially explain our inability to screen FTC. Future studies should incorporate miR-129-1-3p as well as other newly identified miRNAs to screen for FTC.

Despite the high prevalence of specific genetic alterations in thyroid cancer [13], our classifier did not confirm their diagnostic superiority in the screening of thyroid cancer compared to miRNAs. Mutational status was not one of the 25 best markers, and RAS mutation was not retained by our classification algorithm. RAS gene mutations are present in benign and malignant nodules and essentially in follicular lesions, NRAS being the most frequent [28,29]. The risk of malignancy associated with an RAS mutation is variable (31 to 76%), and RAS-mutated benign nodules tend to grow faster than non-RAS-mutated nodules [29], justifying a closer follow-up [28,29].

Mutations are not all equal in predicting malignancy. Some are “strong drivers” (BRAF^V600E^ mutation, RET/PTC rearrangements, etc.) while others are “weak drivers” (RAS, PIK3CA, etc.) [30,31,32]. The utility of miRNA expression in thyroid cancer detection was highlighted by recent studies showing that an miRNA classifier could complement cytology and mutation analysis to identify high risk of malignancy and to predict biological aggressiveness [31,32,33]. Therefore, our signature could be used when “weak drivers” mutations are present to discriminate between high-risk and low-risk FNABs.

Recently, the role of the ThyGeNEXT oncogene panel was used in combination with the expanded miRNA panel ThyraMIRv2 in indeterminate thyroid nodules [34]. This study concluded a reduction in unnecessary surgeries, but surgical cohort was small (n = 90) and the disease prevalence was low (14%). In our study, we included all Bethesda classifications; therefore, it was not possible to obtain the number of unnecessary thyroidectomies. Our study presents several limitations. Our classifier was based on FNABs in ex vivo specimens. With this procedure, the needle enters the nodule directly, without passing through skin, fat and muscle and is therefore less likely to be contaminated by cellular debris. To solve the contamination problem in real conditions, measuring PAX8 mRNA expression to check the proportion of thyroid cells is helpful. We excluded patients for whom localization of the suspicious nodule by palpation was not possible to perform the sampling. Therefore, we excluded nodules smaller than 1 cm or nodules that could not be localized with precision. In these conditions, our cohort was reduced, and our signature was not evaluated on suspicious micronodules (<1 cm). Finally, as already mentioned, while this test is primarily intended for Bethesda III and IV categories, all Bethesda classifications were included in the study. Hence, the number of samples with indeterminate thyroid cytology should be increased.

In the field of medical oncology, the aim of a molecular test is to obtain the greatest NPV to be sure not to miss cancer, while a high PPV will avoid operating the patient when it is not necessary, hence the usefulness of using several diagnostic tests which “complement” each other and refine the screening. The goal of this study was to obtain a signature using new biomarkers, combined or not, that would have predictive values comparable or even superior to the predictive values of the existing molecular tests. Our goal was to improve the positive predictive value (PPV), therefore avoiding unnecessary surgeries (thyroidectomies for benign thyroid tumors).

In the future, it would be interesting to evaluate the feasibility of our test on cytological slides from preoperative FNABs, i.e., in real clinical conditions.

## 5. Conclusions

We developed a molecular signature based on miRNA expression and mutation detection, able to identify malignant nodules with a high PPV and NPV (both 90%), a high specificity (96%) and a competitive sensitivity (76%). This innovative classifier could be effective in improving the screening of thyroid cancer as a complementary test in clinical practice, to reduce the rate of unnecessary surgery and expenses in healthcare. Our data suggest that miRNA expressions emerge as reliable first-line diagnostic markers.

## Figures and Tables

**Figure 1 cancers-16-04214-f001:**
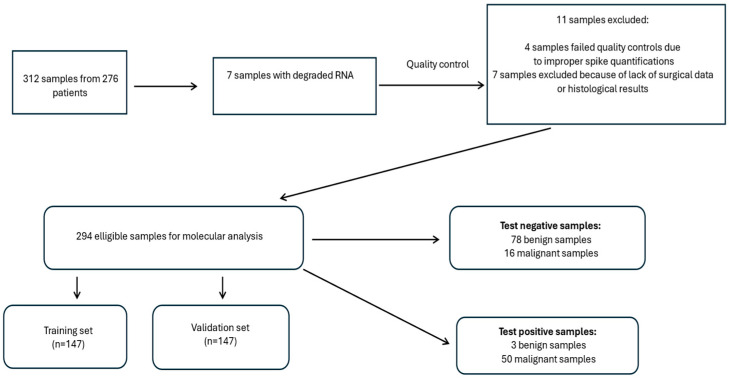
Overview of the study design.

**Figure 2 cancers-16-04214-f002:**
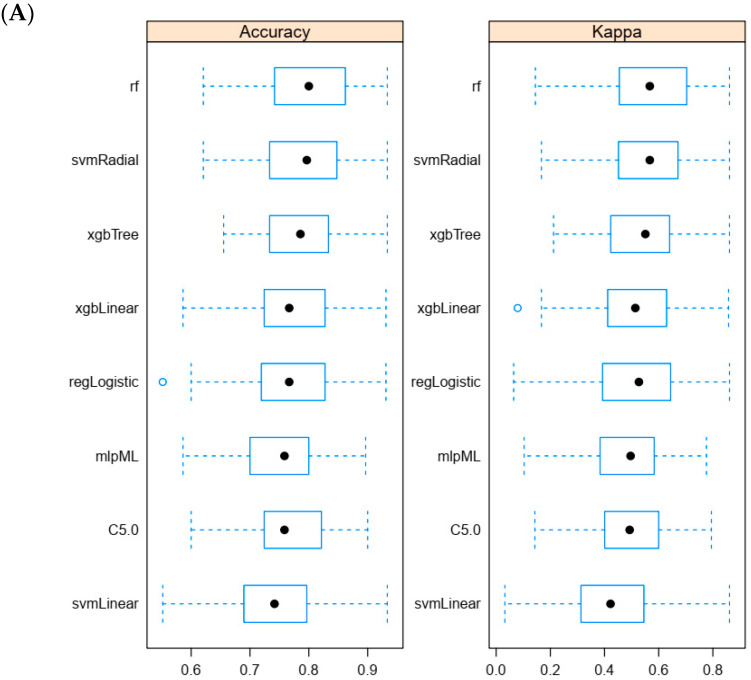
Comparison of the accuracies of the eight different algorithms used to detect malignant nodules. (**A**): Summary of accuracy ranges and Kappa (Cohen) values. (**B**): Receiver Operating Characteristic (ROC) curves and their respective area under the curve (AUC).

**Figure 3 cancers-16-04214-f003:**
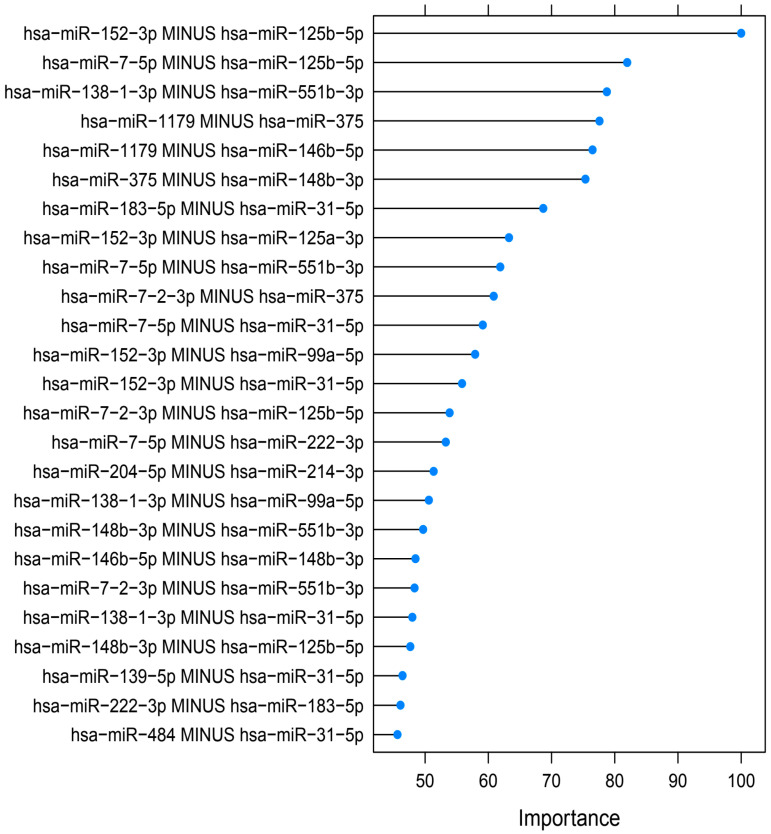
Relative importance score attributed by random forest, showing which miRNA delta pairs (logCt(miRNA1) − logCt(miRNA2)) or gene alterations were the most important to predict the final outcome (benign vs. malignant). All measures of importance were scaled to have a maximum value of 100. Exact calculation methodology can be found on the official documentation of the R Caret package.

**Figure 4 cancers-16-04214-f004:**
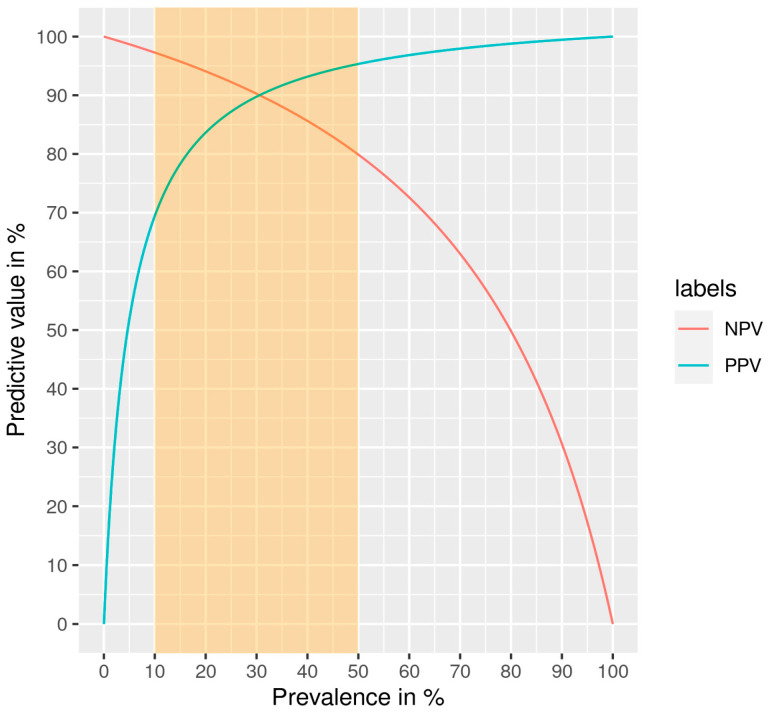
Evolution of the predictive values obtained by the random forest algorithm according to the prevalence of malignancy in our cohort (NPV: negative predictive value, PPV: positive predictive value). The shaded area corresponds to samples for which an indication for thyroidectomy is questionable.

**Table 1 cancers-16-04214-t001:** Characteristics of the samples in the randomized groups.

	Training Set (n = 147)	Validation Set (n = 147)	*p* Overall
Age (median and SD)	52.7 years (17.6)	52.4 years (15.4)	0.879
Female gender	109 (75.2%)	109 (75.2%)	1.000
Male gender	36 (24.8%)	36 (24.8%)	
Bethesda classification			0.936
THY1	6 (5.83%)	5 (4.50%)	
THY2	36 (35.0%)	41 (36.9%)	
THY3	26 (25.2%)	23 (20.7%)	
THY4	18 (17.5%)	19 (17.1%)	
THY5	3 (2.91%)	5 (4.50%)	
THY6	14 (13.6%)	18 (16.2%)	
Pathological diagnosis			0.362
PTC	40 (27.2%)	55 (37.4%)	
FTC	10 (6.80%)	7 (4.76%)	
NIFTP	3 (2.04%)	3 (2.04%)	
PDTC	1 (0.68%)	0 (0.00%)	
MTC	3 (2.04%)	1 (0.68%)	
Benign	90 (61.2%)	81 (55.1%)	
T diagnostic			
T1	25 (50.0%)	33 (61.1%)	
T2	11 (22.0%)	9 (16.7%)	0.521
T3	14 (28.0%)	12 (22.2%)	
N diagnostic			
N0	17 (44.7%)	16 (45.7%)	
N1	21 (55.7%)	19 (54.3%)	1.000

*p* Values were calculated with X^2^ tests for proportions and Mann–Whitney tests for continuous variables (SD: standard deviation, PTC: papillary thyroid carcinoma, FTC: follicular thyroid carcinoma, NIFTP: non-invasive follicular thyroid neoplasm with papillary-like nuclear features, PDTC: poorly differentiated thyroid cancer, MTC: medullary thyroid carcinoma).

**Table 2 cancers-16-04214-t002:** Sensitivity, specificity, PPV (positive predictive value) and NPV (negative predictive value) of our classifier obtained by the random forest (rf) algorithm with the validation set (CI: confidence interval).

Actual			
	Malignant (n = 66)	Benign (n = 81)	
Malignant	50	3	PPV = 0.943 (CI: 0.845–0.981)
Benign	16	78	NPV = 0.83 (CI: 0.761–0.882)
	Sensitivity = 0.758 (CI: 0.636–0.855)	Specificity = 0.963 (CI: 0.896–0.992)	

**Table 3 cancers-16-04214-t003:** Comparison of the performances of recent diagnostic tests.

Diagnostic Test	Total Number of Samples (n)	Bethesda Classification of the Samples	Sensitivity (%)	Specificity (%)	PPV (%)	NPV (%)	Disease Prevalence (%)
Current study	n = 294	I to VI	76%	96%	90%	90%	30%
Thyro Seq v3 [19]	n = 257	III, IV, V	93%	81%	68%	97%	30%
Rosetta GX Reveal [20]	n = 375	II to VI	85%	72%	59%	91%	32%
Multiplatform mutation and miRNA test [21]	n = 109	III and IV	89%	85%	74%	94%	32%
Mir-THYpe test [22]	n = 440	III and IV	89%	82%	66%	95%	29%

## Data Availability

The original contributions presented in this study are included in the article/Appendix A. Further inquiries can be directed to the corresponding author(s).

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
