# Peer review of "Description of a New miRNA Signature for the Surgical Management of Thyroid Nodules"

_cancers, 2024, doi:10.3390/cancers16244214_

Round 1
Reviewer 1 Report
Comments and Suggestions for Authors
In the manuscript titled “Description of a new molecular signature for the surgical management of thyroid nodules based on miRNA expression and mutation detection,” Quiriny and colleagues analyzed miRNA expression and the presence of specific point mutations or mRNA fusions on malignant thyroid nodules samples to propose a new detection strategy. However, the study has several limitations, and the data do not fully support the authors' conclusions. Therefore, some key points that need further discussion are listed below:
Major comments
· The manuscript title, “Description of a New Molecular Signature for the Surgical Management of Thyroid Nodules Based on miRNA Expression and Mutation Detection,” is misleading. According to the authors' data, the mutation and fusion results were not statistically significant; therefore, the title should be revised to reflect the study’s findings more accurately.
· The authors should include graphs illustrating miRNA expression levels, at least for the 25 miRNAs identified as relevant. Currently, the manuscript only presents the Importance Score, which does not provide sufficient insight into the expression data.
· The authors present ROC curves for the best accuracy model using random forest; however, they do not provide the analysis results, only the graphical representation. It is important to include key metrics such as specificity, sensitivity, area under the curve (AUC), accuracy, positive and negative likelihood ratios, and Youden’s index. Additionally, performing a more thorough ROC analysis would facilitate the determination of a cut-off value for miRNA expression, which could be calculated from the available data.
· The comparison of miRNA expression levels in Figure 3 and the supplementary table is unclear. The label “hsa.miR.152.3pMINUShsa.miR.125.5p” is confusing. The authors should clarify why they are comparing these miRNAs against one another instead of using a spike-in control or a housekeeping gene for normalization. This should be explained in the manuscript.
· The authors stratify patient samples into "Training set" and "Validation set." Although this is briefly mentioned in the Data Analysis section (S4), the purpose of separating the patients into these groups is not clearly defined in the manuscript. This should be explicitly clarified in the text.
Minor Comments
- In lines 69-70, abbreviations such as PTC, FTC, and ATC are used without prior explanation. These should be defined before their first use (e.g., Papillary Thyroid Carcinoma [PTC]).
- Figure 1 is unnecessarily large, and the quality of the text is poor. It is recommended to remake the figure, improve its quality, and adjust the first vertical arrow to match the others.
- In the “Supplementary Materials,” Data Analysis (S4), the authors state, “…were manually set to the value of 40 Ct, which we consider to be the maximum theoretical detection value for the qPCR. In other words, we considered the quantities of those miRNA measurements to be practically undetectable.” The phrase “In other words…” onward is unnecessary and redundant, as this concept has already been explained.
- In Table 1, some values are misaligned with their respective row labels. For example, the data for “Pathological Diagnosis” and “Bethesda Classification” in the Validation Set column are misaligned.
- The authors exported data using R, which uses periods instead of dashes in miRNA nomenclature. It is important to correct this in the manuscript, replacing “hsa.miR.152.3pMINUShsa.miR.125.5p” with either “hsa-miR-152-3p/hsa-miR-125-5p” or “hsa-miR-152-3p MINUS hsa-miR-125-5p,” or alternatively, consider creating a table categorizing common miRNAs by type.
- Line 227 should be corrected from “ordre” to “order.”
- In lines 365-367, the authors mention that identifying mutations could provide valuable insights as prognostic markers; however, the study's results do not support this conclusion.
Reviewer 2 Report
Comments and Suggestions for Authors
The manuscript is devoted to the development of a molecular test for the preoperative detection of thyroid cancer. One of the obvious advantages of the study is that the authors used an original set of molecular markers; one of the shortcomings is that the authors did not seem to have decided what exactly their test was needed for.
Section-by-section comments.
Introduction
Line 39: …follicular carcinomas (about 15%)
Do you really have such a percentage of follicular cancer? I have found that there are actually fewer (>5%) of them than is usually reported (10-15%).
Lines 47-49: The diagnosis of malignant thyroid nodules is based on the cytological results of the fine-needle aspiration biopsy (FNAB). The criteria to identify a suspicious nodule are defined in the European Thyroid Imaging and Reporting Data System called EU-TIRADS [3].
The first sentence says about cytology, and then about ultrasound, how does this fit together?
Lines 69-71: Whereas PTC and FTC present up and downregulated miRNAs, ATC show almost exclusively downregulated miRNAs [10], suggesting that miRNA might play a role in the transition from slow progressing differentiated papillary or follicular carcinoma into aggressive thyroid cancers (poorly differentiated and anaplastic thyroid cancer).
Why is there a discussion about anaplastic cancer here if the research does not concern it?
General note: While I understand the purpose and topic of this paper, the Introduction does a poor job of not providing insight into why molecular testing is needed in the preoperative diagnosis of thyroid nodules and what role it should play. This is not strictly necessary, but I recommend rewriting the Introduction to make it clearer and to give a more complete picture of the state of the art in this field.
2. Materials and methods:
2.1. Sample Collection:
The total number of samples should be written here, and even better, the total number of samples with different histological diagnoses.
Lines 102-104: The study was performed following the double-blinded rules: pathologists ignored the molecular analysis results and none of the personnel involved in the molecular analyses were aware of histopathological classifications.
It is not entirely clear about the molecular analysis – how did the classifiers learn if the results of the histological analysis were not known? In general, why is this blinding necessary at the stage of creating a classifier? This makes sense already during test validation.
2.3. Molecular analyses:
Line 113: 34 candidate miRNA biomarkers…
All miRNAs that were used in the study should be listed here.
Lines 114-115: In addition, miR-23a-3p and miR-451a were added to monitor hemolysis.
Further in the article nothing is written about hemolysis control, and these two miRNAs were simply used for classification like the other 34. Therefore, if hemolysis control did exist, then it should be written about it, if not, then just write about 36 candidate miRNAs, and the part about hemolysis can be removed.
Molecular analyses (S2)
The first paragraph, I think, should be moved to the text of the article. The second paragraph, as I understand it, refers to Molecular analyses (S3).
Data analyses (S4):
In this part, we need to explain how many markers the classifiers were trained on. I will describe this issue in more detail in the Results.
In addition, the paragraphs:
“The raw input data were first cleaned…”
“For further analysis, we split the processed dataset…”
Should be moved to the text of the manuscript.
Analytical validation (S5):
Assay precision
What is meant by “precision” here? If it is a measure of the spread of values (e.g. Ct), then the coefficient of variation for markers should be given; if it is reproducibility, then the % of matching results should be given.
2.4. Data analyses
Line 138: Statistical analyses were performed in Excel version.
What does this mean, is it a typo?
Results:
I wrote about this above, but I'll write about it here too – you got 630 markers and I would like to get an explanation, was there a preliminary selection of markers or were they all used to train the classifiers? As I understand it, different methods have different perceptions when the dimension of the sample data vector is greater than the number of samples themselves, for statistical methods this is a particularly bad situation, especially considering that most of these markers are garbage.
Line 202: Among the benign nodules, rf identified…
What is “rf” here?
Since the mutation status was determined for all samples, these results should also be presented, possibly in the Supplementary Materials.
4. Discussion:
The discussion is quite long and detailed, but there are a few topics missing that need to be addressed:
1. What is the ultimate goal for molecular tests – to detect malignant tumors that need to be operated on, or benign ones that can be observed.
2. Based on the above, what NPV and PPV values should be aimed at and why, have these values been achieved?
3. The authors' previous work focused on detecting miRNAs for PTC diagnostics, and some miRNAs were also selected from the literature (by what criteria?) – do the authors think that their test will only detect PTC well? In this work, it turned out that FTC was detected as malignant in half of the cases (maybe these were FV-PTC?), and what about MTC and oncocytic tumors? By the way, the classification results for all samples can be included in the Supplementary Materials.
Round 2
Reviewer 1 Report
Comments and Suggestions for Authors
Dear Author's
I have reviewed your responses to the suggestions raised. Based on the answers provided, I accept the manuscript in its current form.